# ASTROSPLAT: ASSET TRANSFER ORIENTED 3D GAUSSIAN SPLATTING FOR AUTONOMOUS DRIVING

## ABSTRACT

A key component to enable autonomous vehicles (AV) at scale is realistic camera and lidar data simulation for exhaustive validation and testing. To this end 3D Gaussian splatting (3DGS) has gained popularity to simulate camera data due to its high fidelity and rendering speed. A recent work, SplatAD, is the first 3DGS-based method that also renders lidar data in addition to camera data. To capture view-dependent effects, SplatAD uses decoders for camera and lidar renderings that are optimized per scene. However, using scene-specific decoders limits the reusability of the learned Gaussians for the assets across scenes due to scene-specific learned feature representations. Enabling such reusability is crucial to generate rare-event-scenarios at scale for AV stack evaluation and synthetic data creation. Addressing this key limitation, we propose AstroSplat, oriented toward asset transfer across scenes with learned representations that are memory-efficient. Instead of optimizing the decoders per scene, AstroSplat optimizes them per Gaussian enabling high fidelity transfer of assets across scenes. Empirical results across a suite of benchmark datasets and tasks demonstrate that AstroSplat is competitive with prior methods in terms of reconstruction quality, both for camera and lidar renderings. In the asset transfer task, AstroSplat outperforms SplatAD by $10^4\times$ on image generation quality metrics.

## 1 INTRODUCTION

Large-scale testing is essential to ensure the safety of autonomous vehicles (AV) before they can be deployed in the real-world. Simulation allows a scalable approach that allows rapid prototyping of diverse and edge-case scenarios encountered in the real world (Wymann et al., 2000; Dosovitskiy et al., 2017; Caesar et al., 2021; Li et al., 2022; Gulino et al., 2023). However, these simulators are either simplified representations of the world in the form of bounding boxes and HD maps (Caesar et al., 2021; Gulino et al., 2023) or simulate sensor data (camera and lidar) via hand-coded assets (Dosovitskiy et al., 2017; Li et al., 2022). This results in a sim-to-real gap (Pasios & Nikolaidis, 2025) making evaluation of the AV stack unreliable. A key missing piece is high fidelity realistic sensor simulation that can be consumed by the AV stack directly.

To close this gap, a plethora of recent methods have focused on generating photorealistic sensor data. Such methods are primarily based on either neural radiance fields (NeRFs) (Tonderski et al., 2024), 3D Gaussian splatting (3DGS) (Hess et al., 2025), or diffusion (Yang et al., 2024). Current diffusion-based methods (Gao et al., 2023; Yang et al., 2024; Gao et al., 2023; Wang et al., 2024; Zhao et al., 2025; Ni et al., 2025; Mei et al., 2024; Garg & Krishna, 2024) are only limited to simulating camera data. Recent work such as NeuRAD (Tonderski et al., 2024) – based on NeRFs, and a subsequent work SplatAD (Hess et al., 2025) – based on 3DGS, also simulate 360° lidar data along with camera data. Simulating lidar data unlocks the ability to evaluate AV stacks that utilize lidar data for spatial information.

SplatAD outperforms NeuRAD in terms of inference speed and across a majority of camera and lidar data simulation metrics. Although, SplatAD shows impressive results with 360° lidar data, it suffers from a key shortcoming; (**S1**) the camera and lidar decoders are not optimized per Gaussian hence performing poorly in asset transfer between scenes. Asset transfer between scenes is important because it allows efficient simulation, insertion, and manipulation of assets such as vehicles across many driving scenes at inference. This facilitates scalable and diverse scenario generation, including

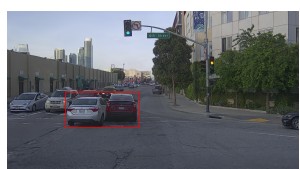 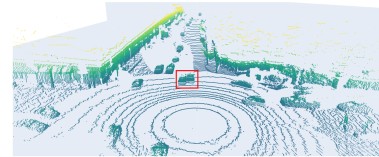

Figure 1: Rare-event collision scenario generated via asset transfer across two different scenes using our proposed AstroSplat. (*Left*) Camera rendering of a target scene. (*Middle*) Camera rendering consisting of an asset (red car) extracted from another scene and inserted into the target scene simulating a collision in the target scene. (*Right*) 360° lidar rendering containing the asset inserted into the target scene.

rare or safety-critical edge cases, by quickly creating realistic sensor-specific (camera/data) data where key assets behave naturally and are visually consistent with their surroundings. Case in point is a rare-event scenario of a collision in the ego lane that we simulate by inserting an asset from another scene not present in the target scene as illustrated in Fig. 1. A method that allows such high-fidelity asset transfer is also crucial for synthetic generation of out-of-distribution scenarios at scale to evaluate (and train) multi-modal fusion methods for perception tasks (Liu et al., 2022; Liang et al., 2022; Li et al., 2024). As shown later in Section 4, SplatAD fails to perform asset transfer reliably.

A possible existing solution that may be used for asset transfer is the use of spherical harmonics (SH) (Müller, 2006) to capture view-dependent effects Kerbl et al. (2023). SH functions are a set of 2D functions defined on the surface of a sphere that can capture effects such as the interaction of light with an object. By assigning weights (SH coefficients) to a set of SHs and adding them together, complex distributions of light and color may efficiently represent complex distributions of color. However, this approach has a shortcoming (**S2**); SH coefficients maybe sparse making them memory inefficient. Later in Section 4 we empirically demonstrate the advantages of learned feature vectors and non-linear decoders to capture view-dependent effects over SH functions.

Addressing shortcomings **S1**–**S2**, we present an **as**set **tr**ansfer **o**riented 3D Gaussian splatting method denoted AstroSplat. AstroSplat is a 3DGS-based method that makes the following novel contributions: (**C1**) introducing learnable non-linear decoders for camera and lidar renderings that are shared across the scene but optimized per Gaussian, and (**C2**) enabling asset transfer across scenes with high fidelity. AstroSplat enables asset transfer across scenes, both in camera and lidar sensor modalities, which to the best of our knowledge hasn't been explored in depth in prior work.

We conclude our study with four sets of experiments: (**G1**) we empirically demonstrate the advantage of learned features and shared non-linear decoders over SHs via sensitivity analysis; (**G2**) we undertake a comparative evaluation by benchmarking AstroSplat against prior work in terms of camera rendering and (**G3**) lidar rendering; (**G4**) we test the effectiveness of AstroSplat in performing asset transfer across scenes. We observe that in terms of camera and lidar renderings, AstroSplat is competitive with respect to the considered baselines. At the same time, AstroSplat is effective in performing asset transfer across scenes addressing the shortcomings of SplatAD.

## 2 RELATED WORK

**GS-based camera rendering for AVs.** A plethora of methods have adapted 3D Gaussian Splatting (3DGS) (Kerbl et al., 2023) for autonomous vehicle (AV) camera rendering. Periodic Vibration Gaussians (PVG) (Chen et al., 2023) extend 3DGS to dynamic scenes via Gaussian flow but lack explicit actor representation, limiting controllability. Street Gaussians (Yan et al., 2023) addresses this with scene decomposition into static backgrounds and rigid objects using bounding boxes, adding temporal variation via Fourier coefficients. OmniRe (Chen et al., 2024b) further models non-rigid actors like pedestrians. All three initialize Gaussians using lidar but rely only on depth supervision, ignoring sensor-specific effects such as intensity variation, ray dropouts, and rolling shutter artifacts.

**GS-based lidar modeling for AVs.** Several works target 360∘ lidar rendering using 3DGS. LiHi-GS (Kung et al., 2024) projects 3D Gaussians to lidar range images with 2D scale compensation but omits sensor artifacts. SplatAD (Hess et al., 2025) unifies camera and lidar rendering while

modeling such characteristics. Lidar-GS (Chen et al., 2024a) focuses only on lidar. UniGaussian (Ren et al., 2024) supports fisheye cameras and lidar intensities but not artifacts. GS-Lidar (Jiang et al., 2025) uses 2D Gaussians with periodic vibrations and spherical harmonics (SH), trading accuracy for memory. Uni-Gaussians (Yuan et al., 2025) instead adopts Gaussian ray tracing (Moenne-Loccoz et al., 2024), achieving better fidelity yet slower performance than rasterization.

**Asset transfer for AVs.** 3DGS-based asset and style transfer methods remain camera-only, with restrictions such as static scenes (Jain et al., 2024; Liu et al., 2024), low inference (Yu et al., 2024a), or supervision (Yu et al., 2025). For AVs, GenAssets (Yang et al., 2025) employs diffusion over NeRF-based latent spaces to reconstruct and complete assets but outputs only camera renderings. R3D2 (Ljungbergh et al., 2025) enables asset transfer and relighting directly on 3D Gaussians, but diffusion-based relighting is slow and does not extend to lidar.

**Alternatives to SHs for 3DGS.** Spherical harmonics, while suitable for transfer, scale poorly in memory as coefficients grow for complex scenes. Alternatives include modified bases (Zhou et al., 2024), SH quantization with MLPs Girish et al. (2024), and shared MLP decoders (Meyer et al., 2025). The latter resembles our approach but struggles with accurate color modeling under some conditions. Notably, none of these frameworks are trained jointly across camera and lidar or evaluated in AV datasets.

## 3 METHOD

We describe each component of our proposed method, AstroSplat. Fig. 2 contains an overview.

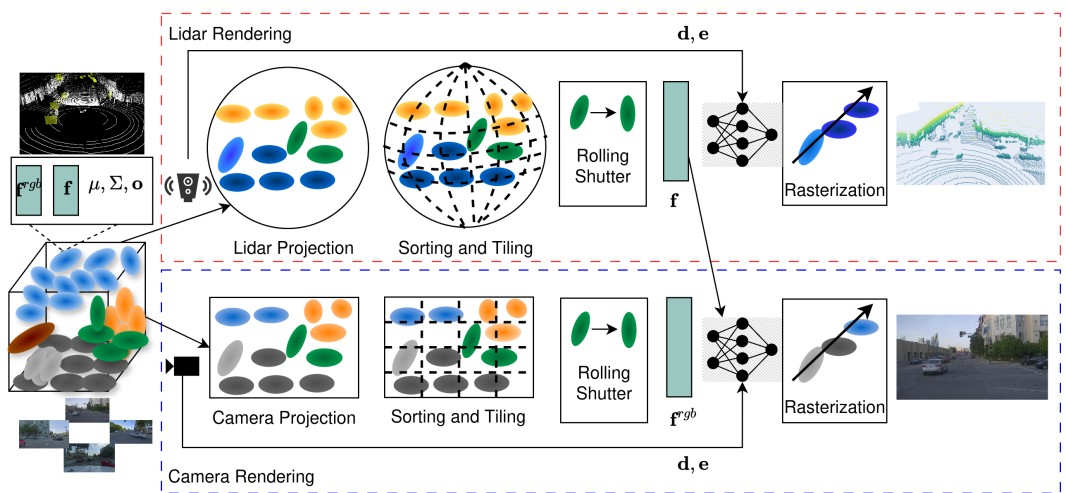

Figure 2: Overview of AstroSplat training. Each 3D Gaussian represented by $\mu, \Sigma, o$ and feature vectors $\mathbf{f}$ and $\mathbf{f}^{rgb}$ is projected onto the camera and lidar sensor modality. The projected Gaussians undergo sensor-specific tiling followed by correction using rolling shutter effects and ego motion. The feature vectors $\mathbf{f}$ and $\mathbf{f}^{rgb}$ are then decoded into RGB values for camera data and intensities, and ray drop probabilities for lidar data using the corresponding MLP decoders. Finally, during rasterization the RGB values, intensities, and ray drop probabilities are $\alpha$–blended producing the camera images and lidar point cloud representations.

**Scene representation.** We use the scene representation from SplatAD (Hess et al., 2025). A scene is described using a collection of 3D Gaussians. The splat model $\mathbf{G}$ is a collection of $N$ Gaussians. Each Gaussian has trainable parameters: opacity $o \in (0, 1)$, mean $\mu \in \mathbb{R}^3$, and an anisotropic covariance matrix $\Sigma \in \mathbb{R}^{3 \times 3}$. To make sure $\Sigma$ is positive semi-definite, it is parameterized by $\Sigma = RSS^T R^T$ where $\mathbf{S} \in \mathbb{R}^3$ is a scale vector and $R \in SE(3)$ is a rotation matrix computed from quaternion $\mathbf{q} \in \mathbb{R}^4$ (Voight, 2021). In addition, every Gaussian is assigned a learnable base color $\mathbf{f}^{rgb} \in \mathbb{R}^3$ and a learnable feature vector $\mathbf{f} \in \mathbb{R}^{D_f}$. The feature vector is used to capture both view-dependent appearance and lidar properties. We also include a trainable embedding $\mathbf{e}$ for each sensor type to reflect sensor-specific appearance differences. $\mathbf{G}$ is represented as, $\mathbf{G} = \left\{ G_i : (\mu_i, \mathbf{S}_i, \mathbf{q}_i, o_i, \mathbf{f}_i^{rgb}, \mathbf{f}_i) | i = 1, \ldots, N \right\}$.

For dynamic scenes, similar to prior methods (Ost et al., 2021; Hess et al., 2025; Kung et al., 2024; Khan et al., 2025; Yan et al., 2023) we adopt a scene graph decomposition, dividing the scene into a static background and multiple dynamic objects. Each moving object is modeled with a 3D bounding box and a sequence of SE(3) poses, which can come from pretrained detectors, trackers, or human annotations. Every Gaussian is given a fixed ID that specifies whether it belongs to the static background or to a particular object. For Gaussians linked to objects, their mean and covariance are expressed in the local coordinates of the object's bounding box. At time $t$, these Gaussians are transformed into world coordinates using the assigned object pose. To handle errors in the pose estimates, we add trainable offset parameters. On top of that, each object has an estimated velocity (based on pose differences) and an additional learnable velocity correction term.

## 3.1 Camera Rendering

Given a camera with a known pose, we combine the set of Gaussians at the specific capture time $t$ and use projection, tiling, and sorting strategies from SplatAD. However, we modify their rasterization and decoding strategy to render an image $I$.

**Projection.** First, each Gaussian's mean and covariance are converted from world coordinates into camera coordinates: $\mu^C = W_W^C \mu, \Sigma^C = W_W^C \Sigma (W_W^C)^T$, where $W_W^C$ is the camera pose with respect to the world frame. $\mu^C$ is then projected to image space as $\mu^I \in \mathbb{R}^2$ using the camera intrinsic matrix $K$, $\mu^I = K\mu^C$. To transform the covariance, we use the top two rows of the projection's Jacobian, so $\Sigma^I = \mathbf{J}^I \Sigma^C (\mathbf{J}^I)^T \in \mathbb{R}^{2\times2}$. Gaussians that do not appear inside the camera's view (frustum) are removed. To do this efficiently, the extent of each Gaussian is approximated by a square axis-aligned bounding box (AABB) covering 99% of its confidence.

**Tiling and sorting.** The image is split into $16 \times 16$ pixel tiles, and every Gaussian is assigned to each tile it overlaps – even if that means duplicating some of them. This ensures that, when rendering, each pixel only processes a limited subset of the total Gaussians. Finally, the Gaussians are sorted by the $z$-depth of their transformed mean $\mu^C$ in the camera view.

**Rolling shutter.** Most cameras use a rolling shutter, capturing images one row at a time. In AV applications, the camera may move during exposure and cause distortions. Accurately modeling effects of rolling shutter is important, especially when the camera moves quickly, because each ray corresponds to a different camera position over time. Instead of projecting every Gaussian to all possible camera poses during exposure - which would be computationally expensive - an efficient approach is to approximate the motion directly in the 2D image space as initially proposed by Seiskari et al. Seiskari et al. (2024) and later also adapted in NeuRAD (Tonderski et al., 2024) and SplatAD. Here, each Gaussian's velocity relative to the camera is used to adjust its projected mean position within the pixel grid based on the capture time for each pixel row, accommodating both static and dynamic scene elements.

For implementation, the pixel velocity for each Gaussian combines the camera's motion and any object dynamics, then this velocity is mapped to image space. When culling Gaussians or checking their overlap with image tiles, their extent is increased to account for their travel during the shutter duration. A rectangular axis-aligned bounding box (AABB) is used, which is enlarged proportionally to the pixel velocity and the rolling shutter time, plus a learnable time offset to correct for differences in sensor timing. This adjustment helps accurately model the area a Gaussian affects during exposure, particularly improving results for narrow or axis-aligned motions, while keeping computational costs reasonable. For in-depth implementation details, the reader is encouraged to refer to SplatAD since we essentially borrow its rolling shutter modeling steps.

**Rasterization.** We now present our main contribution. For each tile and for each Gaussian within the tile, the RGB color $c$ is computed using a small multi-layered perceptron (MLP) decoder shared across all Gaussians in the scene. The feature vectors corresponding to the Gaussian $\mathbf{f}^{rgb}$ and $\mathbf{f}$, the corresponding ray directions $\mathbf{d} \in \mathbb{R}^3$, and a camera-specific learned embedding $\mathbf{e}$ is provided as input to the MLP decoder, i.e., $c = \mathrm{MLP}(\mathbf{f}^{rgb}, \mathbf{f}, \mathbf{d}, \mathbf{e})$. The final step is $\alpha$–blending the RGB values to create an RGB color map in parallel for each tile (and pixel within the tile). For each pixel with coordinates $\mathbf{p} = [p_u, p_v]^T$, the time difference between the pixel's capture time and the image's (of height $H$

and width $W$) middle row is computed using $t_{pix} = (p_v/H - 0.5) \cdot t_{rs}$, where $t_{rs}$ is the rolling shutter duration – the time duration between the last row and first row of the image. $C$ consisting of RGB values at every pixel is computed using:

$$C(\mathbf{p}) = \sum_i^{N_{tile}} c_i \alpha_i(p) \prod_{j=1}^{i-1} (1 - \alpha_j(p)), \tag{1}$$

$$\alpha_i(\mathbf{p}) = \sqrt{\frac{|\Sigma_i^I|}{|\Sigma_i^I + s\mathbf{I}|}} o_i \exp\left(-\frac{1}{2}\Delta_i^T(p)(\Sigma_i^I + s\mathbf{I})^{-1}\Delta_i(p)\right), \tag{2}$$

$$\Delta_i(p) = \mathbf{p} - (\mu_i^I + \mathbf{v}_i^I t_{pix}), \tag{3}$$

where $N_{tile}$ is the sequence of depth sorted Gaussians within a tile, $\Delta_i(p)$ is the $l_1$ distance between the pixel location and Gaussian $i$'s mean $\mu_i^I$ compensated by the rolling shutter – $\mathbf{v}_i^I$ is the pixel velocity, and $s = 0.3$ adopted from Mip-splatting (Yu et al., 2024b) and SplatAD. The RGB color map for the image $I \in \mathbb{R}^{H \times W \times 3}$ is obtained by concatenating the tile-wise color maps.

Note that the rasterization process in AstroSplat is in contrast with SplatAD. SplatAD performs rasterization directly on $\mathbf{f}^{rgb}$ and $\mathbf{f}$ to generate feature maps. To capture view-dependent effects, these feature maps are then passed through a small CNN decoder along with $\mathbf{d} \in \mathbb{R}^{H \times W \times 3}$, and $\mathbf{e}$ to render the image. A major limitation of such a strategy is that the decoder is optimized on the single scene that is trained on. This makes the decoder unsuitable for rendering assets such as vehicles which are essentially a set of 3D Gaussians, trained within a different scene limiting efficient asset transfer. To simulate diverse scenarios at scale, AstroSplat performs the decoding before rasterization, hence optimizing the decoder on every 3D Gaussian instead of the rendered scene enabling high fidelity asset transfer between scenes.

## 3.2 Lidar rendering

Lidar sensors capture spatial information by measuring the distance to different objects in the world to create a discrete set of 3D points (also known as a point cloud) and estimating the material properties of the objects by capturing the amount of light they reflect. Our lidar rendering follows a similar strategy as that of camera rendering. Following SplatAD, we use a non-equidistant tiling strategy. However, similar to the camera rendering we modify the rasterization stage.

**Projection.** Similar to camera rendering, the 3D Gaussians are first transformed from world coordinates to lidar coordinates $\mu^L = [x, y, z]^T$ and $\Sigma^L$, $\mu^L = W_W^L \mu$, $\Sigma^L = W_W^L \Sigma(W_W^L)^T$. Next $\mu^L$ and $\Sigma^L$ are converted from lidar coordinates to spherical coordinates (azimuth $\phi$, elevation $\omega$, and range $r$) to match the standard lidar data capture format. The mean $\mu^S$ is computed as $\mu^S = [r, \phi, \omega] = \left[\sqrt{x^2 + y^2 + z^2}, \arctan 2(y, x), \arcsin(z/r)\right]$, and similar to the projection of $\Sigma^C$ to image space, the covariance $\Sigma^S$ is obtained using $\Sigma^S = \mathbf{J}^S \Sigma^L (\mathbf{J}^S)^T \in \mathbb{R}^{3 \times 3}$, where the Jacobian $\mathbf{J}^S$ is given by

$$\mathbf{J}^S = \begin{bmatrix} \frac{\partial r}{\partial x} & \frac{\partial r}{\partial y} & \frac{\partial r}{\partial z} \\ \frac{\partial \phi}{\partial x} & \frac{\partial \phi}{\partial y} & \frac{\partial \phi}{\partial z} \\ \frac{\partial \omega}{\partial x} & \frac{\partial \omega}{\partial y} & \frac{\partial \omega}{\partial z} \end{bmatrix} = \begin{bmatrix} \frac{x}{r} & \frac{y}{r} & \frac{z}{r} \\ -\frac{y}{x^2+y^2} & \frac{x}{x^2+y^2} & 0 \\ -\frac{xz}{r^2\sqrt{x^2+y^2}} & -\frac{yz}{r^2\sqrt{x^2+y^2}} & \frac{\sqrt{x^2+y^2}}{r^2} \end{bmatrix}.$$

**Tiling and sorting.** Lidar sensors often have uneven vertical (elevation) spacing to improve resolution in specific regions of interest (Hess et al., 2025). Hence, in the tiling process, tiles are designed with a fixed count of diodes vertically and a fixed resolution horizontally (azimuth). Such a strategy avoids the inefficiency of uniform tile sizes by not over-allocating computation where lidar data is sparse. Once assigned to tiles, Gaussians are then sorted by their range $r$.

**Rolling shutter.** Rolling shutter effects for lidar are approximated directly in the spherical coordinate space to simplify computation. The velocity $\mathbf{v}^S$ of each Gaussian is projected into spherical coordinates. Then the first two elements of this velocity are used to enlarge the AABB around each Gaussian by an amount proportional to the rolling shutter period. This adjustment allows for more accurate culling and intersection tests, ensuring that only Gaussians within the sensor's view, and with enough influence, are considered.

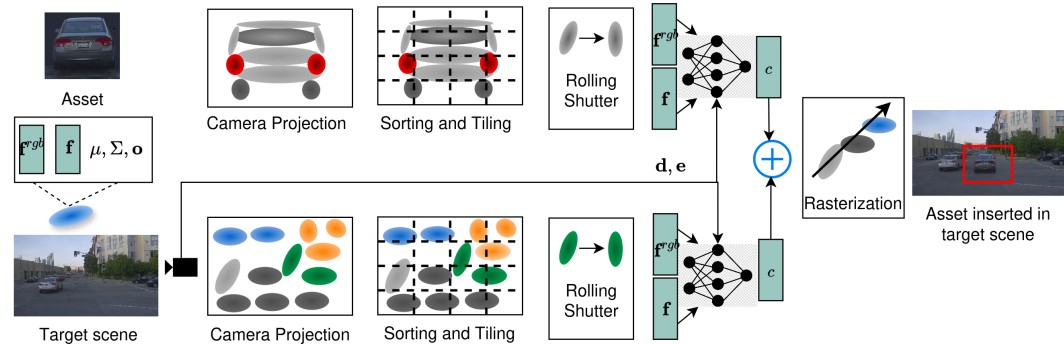

Figure 3: Overview of asset transfer using AstroSplat for camera rendering. An asset extracted from a scene and the RGB values corresponding to it are computed. Similarly RGB values corresponding to a target scene in which the asset will be inserted are computed. The Gaussians and the corresponding colors are then concatenated and $\alpha$–blended together to achieve asset insertion. The asset transfer pipeline is similar for lidar rendering.

**Rasterization.** For each 3D Gaussian, to model the corresponding intensity and ray drop probability (both scalars), $\mathbf{f}$, $\mathbf{d}$, and $\mathbf{e}$ are passed to a small MLP, i.e., $[intensity, ray\_drop] = \text{MLP}(\mathbf{f}, \mathbf{d}, \mathbf{e})$. Then each lidar point within a tile is modeled by $\alpha$–blending the intensities and ray drop probabilities separately following Equations 1, 2, and 3. Following SplatAD and Mip-Splatting (Yu et al., 2024b), $s$ is set such that it corresponds to the geometric mean of the lidar's vertical and horizontal beam divergence. The expected range of a point is obtained by $\alpha$–blending the rolling shutter compensated ranges, i.e., $r_{i,rs} = r_i + \Delta_r^S$, where $\Delta_r^S$ is the distance the lidar and $t_l$ is the time duration of the between the capture of the current lidar point and the middle of the lidar scan. Following SplatAD, we use the expected range for training but the median range during inference. The median range corresponds to the rolling shutter compensated range of the first Gaussian that satisfies $\prod_{j=1}^{i}(1-\alpha_j) < 0.5$.

### 3.3 OPTIMIZATION

We jointly optimize all model components using the loss function proposed in SplatAD. The loss function $L$ is defined as:

$$L = \lambda_r L_1 + (1-\lambda_r)L_{SSIM} + \lambda_{depth}L_{depth} + \lambda_{los}L_{los} + \lambda_{inten}L_{inten} + \lambda_{raydrop}L_{BCE} + \lambda_{MCMC}L_{MCMC},$$

where $L_1$ and $L_{SSIM}$ are $l_1$ are photometric losses the rendered images. $L_{depth}$ and $L_{inten}$ represent $l_2$ losses on the expected lidar range and intensity. $L_{los}$ is a line-of-sight loss that penalizes opacity before the ground truth lidar range. $L_{BCE}$ is a binary cross-entropy loss on the predicted ray drop probability. $L_{MCMC}$ is the regularization term for opacity and scale from Kheradmand et al. (2024).

### 3.4 IMPLEMENTATION AND COMPUTE

For implementation of AstroSplat, we built on top NeuRAD's and SplatAD's official codebases, https://github.com/georghess/neurad-studio and https://github.com/carlinds/splatad respectively. For compute, we used A100 80GB GPUs on which training a scene composed of 100 frames took approximately 45 mins. The total number of GPU-hours used for the reported results and ablation studies was approximately 90 hours.

### 3.5 ASSET TRANSFER

Once we have trained assets, i.e., the set of trained 3D Gaussians, feature vectors, and decoders, AstroSplat facilitates precise and efficient asset transfer across scenes. The Gaussians encode position, geometry, and opacity, the feature vectors encode color and intensity, while the decoders encode view-dependent effects due to changes in lighting in the scene.

As shown in Figure 3, the corresponding Gaussians for the asset(s) from a source scene and the target scene are first projected, followed by tiling and sorting along with rolling shutter compensation. The decoders are then used to decode the view-dependent colors and intensities. The color maps

corresponding to the assets and target scene are then concatenated followed by rasterization which performs $\alpha$–blending.

## 4 EMPIRICAL STUDY

The goals of our reported empirical study are fourfold: (G1) analyze the advantage of learned non-linear decoders over spherical harmonics (SH), (G2) compare AstroSplat against prior work in terms of simulating multi-camera images, (G3) compare AstroSplat against prior work in terms of rendering 360° lidar point clouds, and (G4) evaluate, both qualitatively and quantitatively, the effectiveness of AstroSplat in performing asset transfer across scenes.

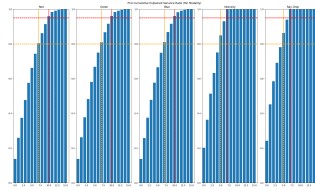

Figure 4: Cumulative explained variance ratio over principal components (per channel) for Pandaset scene "001" (Degree 3 SH).

**Datasets.** Our experiments use two public AV datasets: Pan-daSet (Xiao et al., 2021) and Argoverse2 (Wilson et al., 2023), which vary in lidar and camera configurations, as well as image resolutions. All six PandaSet cameras are used; with Argoverse2, seven ring cameras, excluding black-and-white stereo pairs. Following SplatAD (Hess et al., 2025) and NeuRAD (Tonderski et al., 2024), we evaluate ten challenging sequences per dataset, spanning illumination, dynamic objects, and motion speeds. We use their protocols: full-resolution images and a frame splitting approach for NVS, training on alternate frames and validating on the rest.

**Baselines.** We benchmark AstroSplat versus popular NeRF and 3DGS methods for AVs: UniSim (Yang et al., 2023), NeuRAD (Tonderski et al., 2024), PVG (Chen et al., 2023), Street Gaussians (Yan et al., 2023), OmniRe (Chen et al., 2024b), and SplatAD. We report the quantitative evaluation metrics for all the baselines as presented in SplatAD. Results also include SplatAD with spherical harmonics (SplatAD-SH), which captures view dependency using SHs instead of feature vectors and CNN/MLP decoders. Unlike SplatAD and AstroSplat, which model lidar attributes and view-dependent appearance with shared features, SplatAD-SH provides a degree-0 band for view-independent modeling.

**(G1) Feature representations+MLP decoder vs SHs.** Prior work uses spherical harmonics (SH) for view-dependent appearance, enabling asset transfer between scenes (Yan et al., 2023; Chen et al., 2024b; Khan et al., 2025; Ljungbergh et al., 2025). While SH offers fast inference, it is memory-intensive—optimal SH models require about $3\times$ more parameters than feature splatting (Hess et al., 2025) or AstroSplat. We hypothesize, however, that the SH coefficient space in AV datasets is much lower rank than SH band counts suggest, implying more efficient, realistic alternatives may exist.

To explore this, we perform PCA on SH coefficients from 10 SplatAD-SH models trained on Pandaset validation scenes. We analyze (**a**) intra-channel correlations (within a channel's SH coefficients) and (**b**) inter-channel correlations (across red, green, blue, lidar intensity, ray drop probabilities).

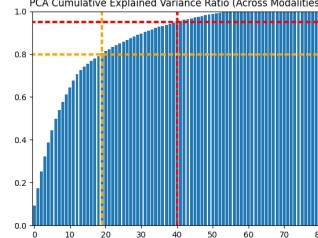

Figure 5: Cumulative Explained variance ratio over principal components (across channels) for Pandaset scene "001" (Degree 3 SH).

For (**a**), we apply PCA per channel. Figure 4 shows for scene "001" that, using 16 SH coefficients (degree 3), RGB channels need just 10 components to explain 95% of variance–suggesting intra-channel compressibility.

For (**b**), PCA on all channel coefficients combined reveals even greater redundancy: as Figure 5 illustrates, 40 components explain 95% and only 19 explain 80% of the variance across 66 SH coefficients.

Together, these experiments suggest that SH coefficients exhibit redundancy and are not optimized for memory consumption. Particularly, the presence of shared information between channels makes our setting a good fit for approaches based on learned per-Gaussian feature representations and a

Table 2: NVS metrics (averaged) for camera and lidar rendering on two datasets. The best performing models on each metric are highlighted; first and second .

| Method | PSNR ↑ | SSIM ↑ | LPIPS ↓ | Depth ↓ | Intensity ↓ | Drop acc. ↑ | CD ↓ |
|---|---|---|---|---|---|---|---|
| | | | **PandaSet** | | | | |
| UniSim | 23.12 | 0.682 | 0.360 | 0.08 | 0.086 | – | 10.3 |
| NeuRAD | 25.80 | 0.753 | 0.250 | 0.01 | 0.063 | 96.2 | 1.9 |
| PVG | 24.01 | 0.712 | 0.452 | 38.74 | – | – | 125.2 |
| Street-GS | 24.73 | 0.745 | 0.314 | 6.18 | – | – | 37.3 |
| OmniRE | 24.71 | 0.745 | 0.315 | 2.88 | – | – | 29.8 |
| SplatAD | 26.76 | 0.815 | 0.193 | 0.01 | 0.059 | 96.7 | 1.6 |
| SplatAD-SH | 24.93 | 0.778 | 0.270 | 0.02 | 0.085 | 96.3 | 1.9 |
| AstroSplat (ours) | 25.51 | 0.805 | 0.240 | 0.01 | 0.065 | 96.5 | 1.6 |
| | | | **Argoverse2** | | | | |
| UniSim | 22.35 | 0.655 | 0.458 | 0.18 | 0.081 | – | 29.2 |
| NeuRAD | 26.18 | 0.721 | 0.310 | 0.02 | 0.058 | 92.2 | 2.6 |
| PVG | 24.47 | 0.712 | 0.510 | – | – | – | – |
| Street-GS | 25.52 | 0.754 | 0.374 | – | – | – | – |
| OmniRE | 25.61 | 0.753 | 0.375 | – | – | – | – |
| SplatAD | 28.42 | 0.826 | 0.270 | 0.02 | 0.052 | 92.6 | 2.8 |
| SplatAD-SH | 25.81 | 0.790 | 0.343 | 0.07 | 0.070 | 93.6 | 4.2 |
| AstroSplat (ours) | 25.71 | 0.800 | 0.320 | 0.05 | 0.055 | 93.3 | 3.7 |

shallow MLP that specify multimodal view-dependent appearance in line with SplatAD and Meyer et al. (2025).

**(G2 & G3) Camera and lidar rendering quality.**
Table 2 presents standard NVS metrics: peak signal to noise ratio (PSNR), structural similarity index metric (SSIM), and learned perceptual image patch similarity (LPIPS) for camera renderings on hold-out validation ego camera poses. AstroSplat is competitive with SplatAD and rest of the considered baselines. This is expected since AstroSplat is not incentivized explicity to outperform SplatAD in NVS metrics. We suspect the CNN decoder in SplatAD to be more effective in capturing texture information as compared to the MLP decoder used in AstroSplat resulting in better NVS metrics.

Table 2 also presents NVS metrics: median squared depth error (Depth), RMSE intensity error (Intensity), ray drop accuracy (Drop acc.), and chamfer distance normalized by the number of ground truth points

Table 1: Inference and resource usage metrics during asset transfer from Pandaset scene "002" to "001". The best performing models on each metric are highlighted; first and second . AstroSplat has the lowest memory footprint and model size.

| | SplatAD-SH | AstroSplat |
|---|---|---|
| **Camera metrics** | | |
| camera_render_time (ms) ↓ | 36.39 | 53.80 |
| camera_gpu_mem (GB) ↓ | 13.14 | 10.27 |
| **Lidar metrics** | | |
| lidar_render_time (ms) ↓ | 228.1 | 230.7 |
| lidar_gpu_mem (GB) ↓ | 15.98 | 14.62 |
| model_size (GB) | 5.4 | 1.3 |

(CD) for lidar renderings on hold-out validation ego lidar poses. AstroSplat is competitive with SplatAD and rest of the considered baselines. Hyperparameter details to reproduce the reported results for SplatAD-SH and AstroSplat have been provided in Appendix A. Appendix C contains an analysis of the sensitivity of AstroSplat to various design choices along with ablation studies.

**(G4) Asset transfer.** We examine the effectiveness of AstroSplat in transferring assets across scenes. For comparison, we analyze how SplatAD fares in this task. In Figure 6, we demonstrate asset transfer by selecting assets from a (source) scene in Pandaset and inserting them into a different (target) scene from Pandaset, qualitatively. SplatAD struggles to render color in the camera renderings faithfully post asset transfer. This is expected as the learned RGB feature representations ($\mathbf{f}^{rgb}$) corresponding to the scenes are scene-specific that possibly have vastly different scene distributions. Quantitatively, the average Fréchet inception distance (FID) scores for the camera renderings after asset transfer across all the frames in selected target scene for SplatAD and AstroSplat is 62.99 and 0.0033 (AstroSplat is $\sim 10^4 \times$ better) respectively. The lidar metric Figure 6 contains lidar renderings after asset transfer using SplatAD and AstroSplat. Point cloud realism metrics like Chamfer distance and earth mover distance don't correctly capture the asset transfer task, thus with only qualitative analysis, we observe that SplatAD fails to model the lidar point cloud faithfully in the presence of occlusions caused by the inserted assets while AstroSplat models them more accurately.

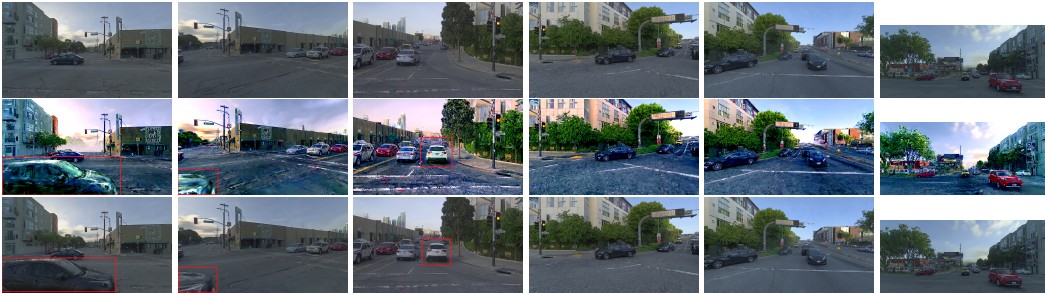

Figure 6: Asset insertion to achieve traffic densification. Multiple assets (black SUV and white car marked in red) are extracted from Pandaset scene "002" (source scene) and inserted into Pandaset scene "001" (target scene). (*Top row*) Multicamera renderings for a frame from the target scene. (*Middle row*) Asset insertion with source and target scenes trained using SplatAD. (*Bottom row*) Asset insertion with source and target scenes trained using AstroSplat. Unlike SplatAD, AstroSplat enables high fidelity asset transfer.

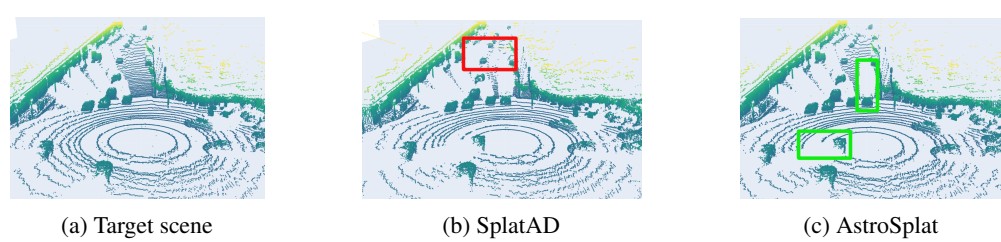

|          (a) Target scene          |          (b) SplatAD          |          (c) AstroSplat          |

Figure 7: Lidar renderings showing asset insertion from source to target scene. (a) Lidar rendering for target scene. (b) SplatAD fails to model occlusions with high fidelity removing points corresponding to the road surface (marked in red). (c) AstroSplat models occlusions accurately (marked in green).

Finally, we report the inference and resource usage metrics for asset transfer from Pandaset scene "002" to "001" using SplatAD-SH and AstroSplat in Table 1. AstroSplat has the lowest memory footprint overall. SplatAD performs decoding on concatenated features during asset transfer to perform rasterization as opposed to the color (and intensity) maps in AstroSplat, hence utilizing higher memory.

## 5 CONCLUSION

We presented AstroSplat, a 3D Gaussian splatting method designed to perform realistic asset transfer for both camera and lidar data across scenes. It addressed a key limitation of SplatAD, a state-of-the-art 3DGS-based method for autonomous vehicles (AVs), of being unsuitable for asset transfer across multiple scenes due to optimizing its feature representations and decoder per scene. AstroSplat optimized the feature representations and decoder per Gaussian instead, enabling high-fidelity transfer of assets. AstroSplat was tested on 2 public datasets for autonomous vehicles (AVs); Pandaset and Argoverse2. We first analyzed the advantage of learning feature representations and a shared decoder over spherical harmonics (SHs) to capture view-dependent color effects in Pandaset. We observed that many SHs coefficients are highly redundant motivating the use of a shallow MLP instead, as incorporated in AstroSplat. We then showed that the rendering quality of AstroSplat for both camera and lidar data is competitive with SplatAD and a host of related prior methods. In the asset transfer task, AstroSplat outperformed SplatAD in the order of $10^4\times$ on standard image quality metrics for the generated camera images. For lidar data, qualitatively we observe AstroSplat to be better at modeling occlusions compared to SplatAD. AstroSplat facilitates easier generation, testing, and evaluation of complex, out-of-distribution scenarios in simulation, helping to safely and efficiently advance AV systems. Future work will explore modeling non-rigid assets such as pedestrians and developing better lighting models to enable asset transfer across a wider variety of scenes with high-fidelity.

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

## A    TRAINING DETAILS

To maintain consistency and fair comparison with SplatAD (Hess et al., 2025), we use the same hyperparameters. Our MLP decoder consists of 2 hidden layers with 32 units in each layer.

## B LLM USAGE

During the development of this paper, Large Language Models (LLMs) were used as an aid for refining the writing and generating scripts for plots. All fundamental research concepts, code architecture, and the overarching framework were created independently by us. LLMs were not involved in generating or shaping the core research ideas.

## C SENSITIVITY ANALYSIS AND ABLATION STUDY

Table 3: NVS metrics averaged over 10 scenes from Pandaset.

| Component | PSNR ↑ | SSIM ↑ | LPIPS ↓ | Depth ↓ | Intensity ↓ | Drop acc. ↑ | CD ↓ |
|---|---|---|---|---|---|---|---|
| Best Model | 25.51 | 0.805 | 0.240 | 0.01 | 0.065 | 96.5 | 1.6 |
| positional_emb_size=10 | 24.95 | 0.784 | 0.259 | 0.02 | 0.073 | 96.5 | 2.2 |
| appearance_emb=False | 24.83 | 0.774 | 0.269 | 0.08 | 0.073 | 96.4 | 2.3 |
| mlp_hidden_units=[32] | 24.80 | 0.775 | 0.270 | 0.03 | 0.074 | 96.5 | 2.3 |

