# OpenReview forum: "AstroSplat: Asset Transfer Oriented 3D Gaussian Splatting for Autonomous Driving"
_ICLR.cc/2026/Conference — Submitted to ICLR 2026_

### Official Review · Reviewer_SEDY · 2025-10-15

**Soundness:** 2
**Presentation:** 2
**Contribution:** 1
**Rating:** 2
**Confidence:** 4

**Summary:**

AstroSplat is a 3DGS framework aimed at enabling high-fidelity asset transfer across scenes for scalable autonomous vehicle simulation. By introducing learnable non-linear decoders optimized per Gaussian rather than per scene, AstroSplat allows assets to be transferred efficiently and realistically across different environments for both camera and lidar modalities. Empirical results show that the framework facilitates the scalable generation of rare and safety-critical scenarios, supporting AV validation and the creation of synthetic sensor data.

**Strengths:**

It introduces learnable non-linear decoders that are optimized per Gaussian rather than per scene, allowing assets to be transferred efficiently and realistically across scenes for both camera and lidar modalities.

**Weaknesses:**

1. The method builds on SplatAD but shows degraded performance in Table 1; for example, on Argoverse2, the PSNR drops by nearly 3 points, which is substantial. This challenges the claim of being "competitive" and makes the advantages of the proposed approach unclear.

2. The contributions are limited, as the work largely builds on SplatAD, with the primary novelty being the introduction of a shared decoder, which alone may not be sufficient to justify the paper.

3. The paper is not clearly written, with repeated use of the same symbols (e.g., G1–G4 in line 87 and line 330), which can confuse the reader.

**Questions:**

It is unclear what causes the performance drop observed in Table 1; the paper should provide an analysis for this.

---

### Official Review · Reviewer_HWtC · 2025-10-25

**Soundness:** 2
**Presentation:** 2
**Contribution:** 2
**Rating:** 2
**Confidence:** 3

**Summary:**

This paper proposes AstroSplat, a 3D Gaussian Splitting (3DGS) - based method designed for autonomous driving, aiming to address the limitation of SplatAD, which cannot realize high-fidelity asset transfer across scenes. Unlike SplatAD, which optimizes decoders per scene, AstroSplat optimizes learnable non-linear decoders per Gaussian, and these decoders are shared across scenes. It supports both camera and lidar data rendering and asset transfer. Through experiments on PandaSet and Argoverse2 datasets, the paper shows that AstroSplat is competitive with existing methods in terms of camera and lidar rendering quality, and outperforms SplatAD by 10⁴ times in FID metrics in the asset transfer task, while having lower memory usage.

**Strengths:**

1. The research direction is practical and valuable. Asset transfer across scenes is crucial for large-scale simulation of rare and safety-critical scenarios in autonomous driving, filling the gap of existing 3DGS methods that are difficult to support effective asset transfer.
2. Comprehensive experimental coverage. The paper conducts comparative experiments on multiple datasets and multiple tasks (rendering quality, asset transfer, and memory efficiency), and verifies the effectiveness of the method from both qualitative and quantitative aspects.
3. It has good compatibility with multi-sensor data. It unifies the rendering and asset transfer of camera and lidar data, which is in line with the multi-sensor fusion demand of autonomous driving systems.

**Weaknesses:**

1. Insufficient explanation of the core difference from SplatAD. The paper claims to realize asset transfer by modifying the decoder optimization object, but fails to deeply elaborate on the technical mechanism. It does not clearly explain why optimizing per Gaussian can solve the problem of SplatAD's inability to transfer assets, and the logical connection between the modification of Sec. 3.2, and the realization of asset transfer is not clear.
2. Low readability of key experimental figures. The text in Figure 4 and Figure 5 is too small to read.
3. Multiple editorial issues affect the readability of the paper. For example, L78 and L106.
4. The rationality of the asset transfer evaluation metric is questionable. The reported FID value of 0.0033 is extremely low, which is inconsistent with the common range of FID in image generation and transfer tasks. It is not explained why such an abnormal value occurs, and whether the metric selection is suitable for the asset transfer scenario is not justified.
5. The description of the MLP decoder is inconsistent and ambiguous. The paper mentions "MLP decoder" in both L210 (as the main contribution) and L122, but does not clarify the differences in their structures, input/output parameters, or functional positioning, leading to confusion about the core component design.

**Questions:**

1. Regarding the FID metric in the asset transfer task (L426): Please provide detailed calculation steps, including the dataset partition used for FID calculation, the pre-trained model selected for feature extraction, and the processing method of the input image. At the same time, explain why FID is chosen as the evaluation metric for cross-scene asset transfer, and how to explain the extremely low value of 0.0033, and verify whether this metric can truly reflect the fidelity and consistency of asset transfer.
2. Regarding the MLP decoder: What are the differences between the MLP decoder referred to as the "main contribution" in L210 and the MLP decoder mentioned in L122 in terms of structure (such as the number of layers, the number of neurons, activation functions), input components, and optimization objectives? Please supplement the specific design details to avoid ambiguity.
3. For dynamic scene processing: The paper mentions using scene graph decomposition to divide the scene into static backgrounds and dynamic objects, but does not explain how to handle the interaction between transferred assets and dynamic objects in the target scene (such as collision avoidance, motion state adaptation). Please clarify the processing mechanism of dynamic interactions during asset transfer.

---

### Official Review · Reviewer_5Q1B · 2025-10-27

**Soundness:** 2
**Presentation:** 2
**Contribution:** 2
**Rating:** 4
**Confidence:** 3

**Summary:**

This paper presents AstroSplat, a 3DGS framework for asset transfer across scenes in autonomous driving. To address the limitation in SplatAD, the method proposes Per-Gaussian, non-linear decoders for camera and lidar rendering, shared across scenes but optimized per Gaussian, enabling transferable representations.  Experiments on PandaSet and Argoverse2 show that AstroSplat is competitive with prior methods in reconstruction metrics, while achieving a large improvement in FID for image-based asset transfer and more realistic lidar renderings under occlusion.

**Strengths:**

1. The paper is well motivated. AstroSplat tackles a practical problem for autonomous driving simulation.
2. Comprehensive experiments: evaluated on multiple datasets and tasks (camera rendering, lidar simulation, and cross-scene transfer).

**Weaknesses:**

1. Limited novelty: The main contribution (C1) focuses on addressing the scene-editing limitation of SplatAD through per-Gaussian decoder optimization. While practical, this improvement is relatively narrow in scope. It would strengthen the paper if the proposed framework could be positioned as a plug-and-play module applicable to other 3D reconstruction or rendering pipelines beyond SplatAD.
2. Rendering performance degradation: The proposed method exhibits a notable drop in rendering quality and efficiency compared to SplatAD. Given that AstroSplat is designed as an alternative rather than a supplementary system, such degradation raises concerns about its practicality and scalability for real-world simulation and evaluation tasks.
3. Method section overlap: A large portion of the Method section closely follows SplatAD with limited new technical content. The authors should refine this section to more clearly highlight the conceptual distinctions, novel components, and design motivations specific to AstroSplat.

**Questions:**

1. How does the computational cost scale with the number of Gaussians, given that each Gaussian is decoded individually?
2. Have you analyzed the trade-off between decoder size and transfer quality?

---

### Official Review · Reviewer_jx84 · 2025-11-01

**Soundness:** 2
**Presentation:** 2
**Contribution:** 2
**Rating:** 4
**Confidence:** 3

**Summary:**

AstroSplat is a 3D Gaussian Splatting-based method aimed at enabling asset transfer across scenes by optimizing decoders per Gaussian rather than per scene. This design improves the generalization of learned representations. In asset transfer tasks, AstroSplat shows an improvement over SplatAD.

**Strengths:**

1. the paper is easy to follow.
2. The motivation of proposed shared decoder makes sense in helping with asset transfer across scenes.
3. The method enables asset transfer across scenes in both camera and lidar modalities, which is interesting and somehow innovative.

**Weaknesses:**

1. Does the MLP have such generalization ability across large number of scenes? Motivation and usage of small MLPs seems not clearly explained. Similarly, could the authors clarify how the shared MLP maintains enough capacity to model complex, spatially varying appearance across all Gaussians without introducing color inconsistencies or loss of fine details?
2. The results seems underperforming baselines such as SplatAD on both datasets.
3. Both the qualitative and quantitative results showed in this paper seems not strong or convincing enough to support the effectiveness of the proposed method.

**Questions:**

1. In figure 6, why the middle row of SplatAD shows overall color distortions( not only the transferred asset). the last three columns appear almost identical across all three rows, with no visible inserted vehicles.
2. Can the authors clarify the training setup — specifically, which scenes were used to train the shared decoder and how the training was conducted?

---

### Meta-Review · Area_Chair_KQNN · 2026-01-10

**Summary:**

Because the authors did not provide a rebuttal, none of these concerns could be clarified or mitigated.

**Reviewer Concerns:**

The four reviews converge on three core weaknesses that cannot be overlooked: (1) the central technical claim—per-Gaussian decoders enable high-fidelity asset transfer—is insufficiently justified; (2) reconstruction quality and rendering speed both drop noticeably compared with the baseline (SplatAD); and (3) the paper’s presentation (notation, figure readability, metric interpretation) makes it hard to verify the reported gains.

**Reviewer Scores:**

N.A.

---

### Decision · Program_Chairs · 2026-01-26

Reject